# A DNA Damage Response Gene Panel for Different Histologic Types of Epithelial Ovarian Carcinomas and Their Outcomes

**DOI:** 10.3390/biomedicines9101384

**Published:** 2021-10-03

**Authors:** Ying-Cheng Chiang, Po-Han Lin, Tzu-Pin Lu, Kuan-Ting Kuo, Yi-Jou Tai, Heng-Cheng Hsu, Chia-Ying Wu, Chia-Yi Lee, Hung Shen, Chi-An Chen, Wen-Fang Cheng

**Affiliations:** 1Department of Obstetrics and Gynecology, College of Medicine, National Taiwan University, Taipei 100226, Taiwan; ycchiang@ntuh.gov.tw; 2Department of Obstetrics and Gynecology, National Taiwan University Hospital, Taipei 100226, Taiwan; stilabry@gmail.com (Y.-J.T.); ascheetah@msn.com (C.-Y.W.); 3Department of Medical Genetics, National Taiwan University Hospital, Taipei 100226, Taiwan; pohanlin01@gmail.com; 4Graduate Institute of Medical Genomics and Proteomics, College of Medicine, National Taiwan University, Taipei 100025, Taiwan; 5Institute of Epidemiology and Preventive Medicine, Department of Public Health, National Taiwan University, Taipei 100025, Taiwan; tbenlu@gmail.com; 6Department of Pathology, College of Medicine, National Taiwan University, Taipei 100225, Taiwan; pathologykimo@gmail.com; 7Graduate Institute of Clinical Medicine, College of Medicine, National Taiwan University, Taipei 100225, Taiwan; b101092037@gmail.com (H.-C.H.); plzfixthecar@gmail.com (C.-Y.L.); shkt0802@gmail.com (H.S.); 8Department of Obstetrics and Gynecology, National Taiwan University Hospital, Hsin-Chu Branch, Hsin-Chu 30059, Taiwan; 9Graduate Institute of Oncology, College of Medicine, National Taiwan University, Taipei 100025, Taiwan

**Keywords:** epithelial ovarian cancer, DNA damage response, somatic mutation, clear cell carcinoma

## Abstract

DNA damage response (DDR) is important for maintaining genomic integrity of the cell. Aberrant DDR pathways lead to accumulation of DNA damage, genomic instability and malignant transformations. Gene mutations have been proven to be associated with epithelial ovarian cancer, and the majority of the literature has focused on *BRCA*. In this study, we investigated the somatic mutation of DNA damage response genes in epithelial ovarian cancer patients using a multiple-gene panel with next-generation sequencing. In all, 69 serous, 39 endometrioid and 64 clear cell carcinoma patients were enrolled. Serous carcinoma patients (69.6%) had higher percentages of DDR gene mutations compared with patients with endometrioid (33.3%) and clear cell carcinoma (26.6%) (*p* < 0.001, chi-squared test). The percentages of DDR gene mutations in patients with recurrence (53.9 vs. 32.9% *p* = 0.006, chi-squared test) or cancer-related death (59.2 vs. 34.4% *p* = 0.001, chi-squared test) were higher than those without recurrence or living patients. In endometrioid carcinoma, patients with ≥2 DDR gene mutations had shorter PFS (*p* = 0.0035, log-rank test) and OS (*p* = 0.015, log-rank test) than those with one mutation or none. In clear cell carcinoma, patients with ≥2 DDR gene mutations had significantly shorter PFS (*p* = 0.0056, log-rank test) and OS (*p* = 0.0046, log-rank test) than those with 1 DDR mutation or none. In the EOC patients, somatic DDR gene mutations were associated with advanced-stage tumor recurrence and tumor-related death. Type I EOC patients with DDR mutations had an unfavorable prognosis, especially for clear cell carcinoma.

## 1. Introduction

Epithelial ovarian carcinoma (EOC) is a major cause of death in women worldwide, and patients are usually diagnosed at an advanced stage with a 5-year survival of less than 50% [1,2,3,4]. Clinical prognostic factors include cancer stage, histological subtypes, tumor grade, residual tumor size after debulking surgery and response to chemotherapy. Despite an initial good response to primary treatments of debulking surgery and adjuvant platinum-based chemotherapy, the majority of patients experience a cancer relapse that is resistant to salvage treatments and eventually die of the disease [4,5].

Precision medicine is the current direction for cancer management depending on the specific genetic or molecular features of cancer. There are several subtypes of EOC—high-grade serous, clear cell, endometrioid, mucinous and low-grade serous—that could be viewed as distinct diseases for their differences in clinical course and pathological features [6,7]. To date, the most promising target therapies for EOC are anti-angiogenesis agents and poly ADP-ribose polymerase inhibitors (PARPi). Bevacizumab in combination with chemotherapy has demonstrated improved progression-free survival, and an overall survival benefit in high-risk patients [8,9,10]. Maintenance therapy with PARPi has revised the management of EOC in newly diagnosed and recurrent diseases. The identification of *BRCA* mutations or homologous recombination deficiency (HRD) status is critical for selecting potential patients, but both positive and negative patients as defined by current HRD assays benefited from PARPi [11,12,13,14,15].

DNA damage response (DDR) is important for maintaining a cell’s genomic integrity, and the DDR pathway is composed of various molecules that detect DNA damage, activate cell-cycle checkpoints, trigger apoptosis, and coordinate DNA repair [16,17,18]. Several exogenous or endogenous sources (e.g., oxidative damage, radiation, ultraviolet light, cytotoxic materials, replication errors) may result in DNA damage that may eventually lead to genomic instability and cell death [19]. DDR consists of several pathways, including base excision (BER), mismatch (MMR) and nucleotide excision repair (NER); translesion synthesis (TLS) for single-strand break repair; homologous recombination (HR) and nonhomologous DNA end joining (NHEJ) for double-strand break repair; and cell cycle regulation (CCR) (27, 28). Homologous recombination is an error-proof repair pathway to restore the original sequence at the double-strand DNA break. *BRCA* 1/2 genes participating in HR and maintaining PARPi therapy for *BRCA*-mutated EOC is a good example of synthetic lethality [20]. Several other DDR genes have been identified as potential targets for novel cancer therapy under clinical investigation [16,17]. Understanding the complex DDR pathways is helpful for exploring the feasibility of novel DDR inhibitors in clinical practice. In the study, we investigated the somatic mutations of DDR genes in 172 EOC patients using a targeted DDR gene panel using a next-generation sequencing method. The correlation of the somatic DDR gene mutations, clinical parameters and outcomes was analysed.

## 2. Materials and Methods

### 2.1. Patients and Specimens

The study protocol was approved by the National Taiwan University Hospital Research Ethics Committee (201509042RINA, approved on 24 November 2015 and 201608025RINA, approved on 07 October 2016). Informed consent from all participants was obtained and the methods were performed in accordance with the guidelines and regulations. From December 2015 to October 2018, 172 women diagnosed with epithelial ovarian cancer who had received debulking surgery and adjuvant chemotherapy were enrolled. The cancerous tissue specimens collected during debulking surgery were immediately frozen in liquid nitrogen and stored at −70 °C. A portion of the tissue specimens were sent for pathological examinations to confirm the diagnosis and ensure tumorous tissue sufficient for the following experiments. Clinical data were obtained from medical records, including age, cancer stage, the findings during debulking surgery, treatment course and recurrence. Optimal debulking surgery was defined as a maximal residual tumor size <1 cm following surgery. The tumor grade based on International Union Against Cancer criteria, and cancer stage was based on International Federation of Gynecology and Obstetrics (FIGO) criteria [21]. All patients received platinum-based adjuvant chemotherapy and regular follow-ups after primary treatments. Recurrence was defined as abnormal results from imaging studies (including computerized tomography or magnetic resonance imaging), elevated CA-125 (more than twice the upper normal limit) for two consecutive tests in 2-week intervals, or a biopsy-proven disease. Progression-free survival (PFS) was defined as the time from the date of primary treatment completion to the date of confirmed recurrence, disease progression or last follow-up. Overall survival (OS) was defined as the period from surgery to the date of death related to EOC or the date of last follow-up.

### 2.2. The Panel of DNA Damage Repair Genes

We selected 60 genes involved in DNA damage response (DDR) for the gene panel (Table 1), including genes of homologous recombination (HR), nonhomologous DNA end joining (NHEJ), base excision repair (BER), mismatch repair (MMR), nucleotide excision repair (NER), translesion synthesis (TLS) and cell cycle regulation (CCR) [16,17].

### 2.3. Genomic DNA Extraction

Genomic DNA was isolated using a QIAGEN Genomic DNA extraction kit according to the manufacturer’s instructions (Qiagen Inc., Valencia, CA, US). The purity and concentration of the genomic DNA were checked by agarose gel electrophoresis and the OD_260/280_ ratio.

### 2.4. Library Preparation, Next-Generation Sequencing, and Sequence Mapping

The genomic DNA was fragmented with Covaris fragmentation protocol (Covaris, Inc., Woburn, MA, US). The size of the fragmented genomic DNA was checked by Agilent Bioanalyzer 2100 (Agilent Technologies, Inc., Santa Clara, CA, US) and NanoDrop spectrophotometer (Thermo Fisher Scientific, Inc., Wilmington, DE, US). The target gene library was generated with NimblGen capture kits (Roche NimblGen, Inc. Hacienda Dr Pleasanton, CA, US). The samples were sequenced by Illumina MiSeq with paired-end reads of 300 nucleotides.

The analysis algorithm was conducted according to our previous protocol [22]. Briefly, the raw sequencing data were aligned with the reference human genome (Feb. 2009, GRCh37/hg19) with Burrows–Wheeler Aligner software (version 0.5.9) [23]. SAM tools (version 0.1.18) was used for data conversion, sorting, and indexing [24]. For single nucleotide polymorphisms (SNPs) and small insertion/deletions (indels), Genome Analysis Toolkit (GATK; version 2.7) was used for variant calling with Base/indel-calibrator and HaplotypeCaller. Pindel or Breakdancer software were used for structural variants larger than 100 bp which cannot be identified by GATK, such as large deletions, insertions and duplications [25]. After variant calling, ANNOVAR was used for annotation of the genetic variants [26,27]. The dbSNP, Exome sequencing Project 6500 (ESP6500) and the 1000 Genomes variant dataset were used to filter common variants of sequencing results.

### 2.5. Variant Classification

The sequence variants were classified according to the IARC variant classification [28]. The pathogenic mutations were defined as large-scale deletion, frame-shift mutation, nonsense mutation, genetic variants associated with uncorrected splicing and mutations affecting protein function demonstrated by functional analyses. The pathogenic and likely pathogenic mutations were used as deleterious mutations in our study. An allele frequency greater than 0.01 in the general population in the 1000 Genomes variant dataset or ESP6500 database were considered benign or likely benign genetic variants. Silent and intronic variants that did not affect splicing were also considered benign or likely benign. Other variants, mainly missense mutations without known functional data, were considered as variants of uncertain significance (VUSs). To reduce their number, bioinformatics analyses, including PolyPhen2 and SIFT, were used to evaluate potential pathogenicity [29,30,31]. The VUSs were suspected of being deleterious mutations if they met two criteria: (1) a population frequency of less than 0.01 in the 1000 Genomes and ESP6500 databases and (2) a bioinformatics analysis result with a SIFT score less than 0.05 and a polyphen2 score greater than 0.95.

### 2.6. Statistical Analysis

All statistical analyses were performed using the Statistical Package for Social Sciences software package (IBM SPSS Statistics for Windows, Version 22.0. IBM Corp. Armonk, NY, US) and R (version 3.1.2, The R Foundation for Statistical Computing, Institute for Statistics and Mathematics, Wirtschaftsuniversität Wien, Welthandelsplatz Vienna, Austria). One-way ANOVA was used to compare continuous variables and a chi-squared test was used for categorical variables. Survival curves were generated using the Kaplan–Meier method, and differences were calculated using the log-rank test. A multivariate Cox’s regression model was used to evaluate the prognostic factors for progression-free survival (PFS) and overall survival (OS). Statistical significance was set as a *p* value of less than 0.05.

## 3. Results

### 3.1. Clinical Characteristics of the Patients

There were 172 EOC patients enrolled: 69 serous, 39 endometrioid and 64 clear cell carcinomas (Table 2). There were 68 high-grade serous carcinomas (type II tumor) and 104 type I tumors. The median age was 52, and the median pre-treatment CA125 value was 400 U/mL; 59.9% were diagnosed at an advanced cancer stage, and 65.1% had undergone optimal debulking surgery; 59.3% had disease recurrence, and 44.2% died of EOC. All patients received adjuvant platinum and paclitaxel chemotherapy.

### 3.2. Deleterious DDR Gene Mutations

As shown in Table 3, 114 deleterious somatic mutations were identified from 26 genes of our 60-gene DDR panel in 78 EOC patients: 27 nonsense mutations in 23 patients, 28 frameshift mutations in 20, 28 missense mutations in 26 patients and 31 mutations involving uncorrected splicing in 29 patients. There were single-gene mutations in 57 patients, and multiple-gene mutations in 21: 2 mutations in 14 patients, 3 mutations in 2, 4 mutations in 3, 5 mutations in 1 and 6 mutations in 1 patient (Figure 1). We also identified 109 missense mutations classified as variants of uncertain significance (VUSs) with the potential of being deleterious mutations after searching the database (http://www.ncbi.nlm.nih.gov/snp, accessed on 28 September 2021) and bioinformatic analyses (Appendix A and Appendix A).

The pattern of prevalent mutated DDR genes was different among the histological subtypes (Figure 1). The proportion of wild type DDR genes was 54.7% in all EOC patients; 30.4% in serous carcinoma, 66.7% in endometrioid carcinoma and 73.4% in clear cell carcinoma. The top three prevalent mutated DDR genes were *TP53* (27.9%), *MUTYH* (6.4%) and *BRCA2* (5.8%) for all patients. Serous carcinoma—*TP53* (56.5%), *BRCA2* (5.8%) and *RAD51C* (5.8%); endometrioid carcinoma—*TP53* (15.4%), *ATM* (12.8%) and *MSH2* (7.7%); clear cell carcinoma—*MUTYH* (9.4%), *TP53* (4.7%), *BRCA2* (3.1%) and *ERCC8* (3.1%). The top three prevalent mutated subgroups of DDR genes were CCR (30.8%), HR (10.5%) and BER (7.0%) for all patients. Serous carcinoma—CCR (58.05%), HR (15.9%) and BER (5.8%); endometrioid carcinoma—CCR (23.1%), MMR (15.4%) and HR (7.7%); clear cell carcinoma—BER (9.4%), CCR (6.3%) and HR (6.3%). For detailed information, please refer to Appendix A and Appendix A.

### 3.3. Correlation of DDR Gene Mutations with Clinical Outcomes of the EOC Patients

We evaluated the correlations between the mutation of DDR genes, the clinicopathologic parameters and outcome of the EOC patients. As shown in Table 4, type II tumors had a higher percentage of HR gene mutations than type I tumors (16.18 vs. 6.73%, *p* = 0.048, chi-squared test). Endometrioid carcinoma (15.38%) had a higher percentage of MMR mutations than those of serous carcinoma (2.90%) and clear cell carcinoma (4.69%) (*p* = 0.03, chi-squared test). Low-grade tumors had a higher percentage of MMR mutations compared with high-grade tumors (17.24 vs. 4.20%, *p* = 0.009, chi-squared test). Type II tumors had a higher percentage of DSBR mutations than type I tumors (17.65 vs. 6.73%, *p* = 0.026, chi-squared test). Serous carcinoma (57.97%) had a higher percentage of CCR mutations than those of endometrioid carcinoma (23.08%) and clear cell carcinoma (6.25%) (*p* < 0.001, chi-squared test). Type II tumors had higher percentage of CCR mutations than those of type I tumors (58.82 vs. 12.50%, *p* < 0.001, chi-squared test). The advanced-stage patients had a higher percentage of CCR mutations than the early-stage patients (42.72 vs. 13.04%, *p* < 0.001, chi-squared test). The recurrent patients had a higher percentage of CCR mutations than those without recurrence (39.22% vs. 18.57%, *p* = 0.004, chi-squared test). Patients who died of EOC had higher percentages of CCR mutations than living patients (40.79 vs. 22.92%, *p* = 0.012, chi-squared test). Serous carcinoma (69.57%) had higher percentage of DDR mutations than those of endometrioid carcinoma (33.33%) and clear cell carcinoma (26.56%) (*p* < 0.001, chi-squared test). Type II tumors had a higher percentage of DDR mutations than type I tumors (70.59 vs. 28.85%, *p* < 0.001, chi-squared test). The advanced stage patients had higher percentage of DDR mutations than the early-stage patients (57.28 vs. 27.54%, *p* < 0.001, chi-squared test). Recurring patients had a higher percentage of DDR mutations than those without recurrence (53.92 vs. 32.86%, *p* = 0.006, chi-squared test). Patients who died of EOC had a higher percentage of DDR mutations than living patients (59.21 vs. 34.38%, *p* = 0.001, chi-squared test).

EOC patients without DDR gene mutation had longer progression-free survival (PFS) (*p* = 0.0072, log-rank test, Figure 2A) and overall survival (OS) (*p* = 0.022, log-rank test, Figure 2B) than those with 1 DDR or ≥2 DDR mutations. In serous carcinoma, patients with or without DDR mutations had similar PFS (*p* = 0.56, log-rank test, Figure 2C). Patients with ≥2 DDR mutations had a trend of better OS than those with 1 mutation or none, but it was not statistically significant (*p* = 0.47, log-rank test, Figure 2D). In endometrioid carcinoma, patients with ≥2 DDR gene mutations had shorter PFS (*p* = 0.0035, log-rank test, Figure 2E) and OS (*p* = 0.015, log-rank test, Figure 2F) than those with 1 mutation or none. In clear cell carcinoma, patients with ≥2 DDR gene mutations had significantly shorter PFS (*p* = 0.0056, log-rank test, Figure 2G) and OS (*p* = 0.0046, log-rank test, Figure 2H) than those with 1 DDR mutation or none.

Tumor recurrence with CCR gene mutation (HR: 1.68 (1.12–2.50), *p* = 0.011), 1 DDR gene mutation (HR: 1.71 (1.12–2.60), *p* = 0.013), endometrioid carcinoma (HR: 0.17 (0.08–0.37), *p* < 0.001), type II tumor (HR: 2.69 (1.81–4.00), *p* < 0.001), advanced-stage carcinoma (HR: 5.29 (3.16–8.85), *p* < 0.001), high-grade tumor (HR: 5.57 (2.26–13.70), *p* < 0.001) and optimal debulking surgery (HR: 0.28 (0.18–0.41), *p* < 0.001) were significant in the univariate Cox regression model (Table 5). Advanced-stage carcinoma (HR: 3.08 (1.63–5.80), *p* = 0.001) and optimal debulking surgery (HR: 0.51 (0.32–0.80), *p* = 0.004) were important prognostic factors in the multivariate analysis. Cancer-related death with TLS gene mutation (HR: 33.76 (3.95–289.00), *p* = 0.001), 1 DDR gene mutation (HR: 1.96 (1.20–3.20), *p* = 0.007), endometrioid carcinoma (HR: 0.12 (0.04–0.38), *p* < 0.001), type II tumor (HR: 1.88 (1.19–2.96), *p* = 0.007), advanced-stage carcinoma (HR: 6.84 (3.28–14.25), *p* < 0.001), high-grade tumor (HR: 17.97 (2.50–129.29), *p* = 0.004) and optimal debulking surgery (HR: 0.26 (0.16–0.41), *p* < 0.001) were significant in the univariate Cox regression model. Type II tumor (HR: 0.35 (0.20–0.60), *p* < 0.001), TLS gene mutation (HR: 9.57 (1.08–84.83), *p* = 0.042), advanced-stage carcinoma (HR: 4.82 (2.09–11.09), *p* < 0.001) and optimal debulking surgery (HR: 0.38 (0.22–0.64), *p* < 0.001) were important prognostic factors in the multivariate analysis.

## 4. Discussion

Our study showed that nearly half of the epithelial ovarian cancer (EOC) patients had DNA damage response (DDR) gene mutations with varied proportions of histological subtypes. Two-thirds of serous adenocarcinoma patients, one-third of endometrioid adenocarcinoma patients and one-fourth of clear cell carcinoma patients had DDR gene mutations. Our DDR gene panel consisted of the genes involved in single-strand break repair, double-strand break repair and cell cycle regulation, including the genes recommended by National Comprehensive Cancer Network (NCCN) guidelines as cost-effective tools for assessing the lifetime risk of EOC, such as *ATM, BRCA1/2, BRIP1, MLH1, MSH2, MSH6, PALB2, RAD51C* and *RAD51D* [32]. The major components of DDR gene mutations were CCR in serous, CCR and SSBR in endometrioid and SSBR in clear cell carcinomas; CCR and DSBR in type II tumors (high-grade serous carcinoma in the cohort); and SSBR in type I tumors. A multiple DDR gene panel increased the detection rate of somatic mutation of genes involved in DNA damage repair pathway in comparison with a *BRCA* test alone. The percentage of *BRCA* 1/2 somatic mutation in serous carcinoma was 7.2, which was compatible with the 6–7% in previous studies [33,34,35,36,37]. The non-*BRCA* HR somatic mutation of our study was more than 10% in serous and endometrioid carcinomas, and the MMR somatic mutation was around 15% in endometrioid carcinomas, which was compatible with the previous study [38].

Our study showed that ovarian clear cell carcinoma patients with DDR gene mutations had an unfavorable survival prognosis. Those who had somatic DDR mutations were significantly associated with advanced-stage carcinomas, tumor recurrence and tumor-related death. The trend was different in histological subtypes as serous carcinomas or type II tumors with DDR mutation showed a better survival trend. Non-serous or type I EOC patients with DDR mutations had a poor prognosis, especially in clear cell carcinoma. Ovarian clear cell carcinoma is an aggressive drug-resistant subtype of EOC in association with endometriosis and glycogen accumulation. It accounts for about 5–13% of all EOCs in Western populations, but up to 20–25% in East Asia, including Taiwan [2,3]. Previous studies showed that the somatic mutations of ovarian clear cell carcinoma (mainly in *ARID1A*, *PIK3CA*, *KRAS* and *PPP2R1A*) might be related to chromatin remodeling, cell proliferation, cell cycle checkpointing and cytoskeletal organization [39,40,41,42,43,44,45,46,47,48,49]. However, the frequent mutations of *ARID1A, PIK3CA, PPP2R1A* or *TP53* in ovarian clear cell carcinoma did not correlate well with the prognosis [45]. Other infrequent gene mutations of clear cell carcinoma included *ARID1B, ARID3A, CREBBP, CSMD3, CTNNB1, LPHN3, LRP1B, MAGEE1, MLH1, MLL3, MUC4, PIK3R1, PTEN* and *TP53* [41,43,46,48,49]. DDR gene mutations in ovarian clear cell carcinoma was unclear in the literature, and our finding of an unfavorable prognosis in clear cell carcinoma patients with DDR gene mutations could provide useful information.

Our DDR gene panel could provide a scientific rationale for patient selection in future clinical trials that target DNA damage repair response pathways, especially in clear cell carcinoma. *BRCA* gene tests or companion HRD assays are currently suggested for PARPi, but there are unmet problems that need to be resolved [11,12,13,14,15,20]. The most important one is that the HRD assays cannot consistently identify patients who do not benefit from PARPi therapy. The consensus for the cut-off value was indeterminate because the thresholds of HRD assays were developed from retrospective exploratory analyses [11,50,51]. Generally, advance-stage, high-grade serous carcinoma patients with tumor *BRCA* (t*BRCA*) mutations, including germline (g*BRCA*) or somatic (s*BRCA*), derived the greatest benefit from PARPi maintenance therapy [11,12,13,14,15]. Approximately 11–18% of patients had a g*BRCA* mutation, and another 6–7% patients had an s*BRCA* mutation with a negative g*BRCA* test [33,34,35,36,37]. However, about 5% of g*BRCA* mutated patients tested negative for t*BRCA* [52,53,54]. The non-*BRCA* HR gene mutations were usually pooled together to interpret the association with clinical outcomes in previous studies because of their relatively low prevalence [35,55,56,57]. Twenty-one platinum-sensitive recurrent patients with non-*BRCA* somatic mutations (*BRIP1*, *CDK12*, *RAD54L* and *RAD51B*) derived benefit from olaparib in study 19 [58]. In ARIEL2, there were 20 patients with non-*BRCA* HR gene mutations (*ATM*, *BRIP1*, *CHEK2*, *FANCA*, *FANCI*, *FANCM*, *NBN*, *RAD51B*, *RAD51C* and *RAD54L*), but the sensitivity in discriminating a rucaparib response was only 11% [59]. However, *BRCA* wild type EOC patients still benefitted from PARPi, which indicated that a *BRCA* test by itself was inadequate for selecting EOC patients for PARPi [13,14,15]. It needs to be determined which individual or panel of non-*BRCA* HR genes could be used to predict a PARPi response, especially in non-serous EOC patients.

There were limitations to our study. First, germline gene mutations were not investigated. These not only inform the patients but also identify family members of the possible risk of malignancy [52,53,54]. The NCCN suggested germline gene tests of *ATM, BRCA1/2, BRIP1, MLH1, MSH2, MSH6, PALB2, RAD51C, RAD51D* and *STK11* to assess the lifetime risk of EOC [32], but how many genes should be included in the panel is inconclusive. Second, the numerous variants of uncertain significance (VUSs) identified by multiple gene panels would cause controversy in risk assessment and management [60,61,62]. The biological functions and clinical impacts of most individual mutations in the genomic loci have not been well characterized, especially for VUSs [63]. Even in the well-studied *BRCA* gene, there is a difference among laboratories in the VUS reporting rate (3–50%), detection protocols and management strategies [64]. Further sharing and integration of gene sequencing data in an open database might decrease VUSs. Third, the cohort sample was not large enough; only the trends of clinical prognosis that correlated with each DDR pathway were found. Further large-scale investigations are needed.

## 5. Conclusions

Our study found that nearly half of the EOC patients had DDR gene mutations of varying proportions in the histological subtypes. Patients with somatic DDR mutations were significantly associated with advanced-stage carcinoma, tumor recurrence and tumor-related death. Type I EOC patients with DDR mutations had an unfavorable prognosis, especially for clear cell carcinoma. A broad multiple-gene DDR panel would provide not only comprehensive information of gene mutations but also a rationale for a future study of a novel therapy target for DNA damage response pathways.

## Figures and Tables

**Figure 1 biomedicines-09-01384-f001:**
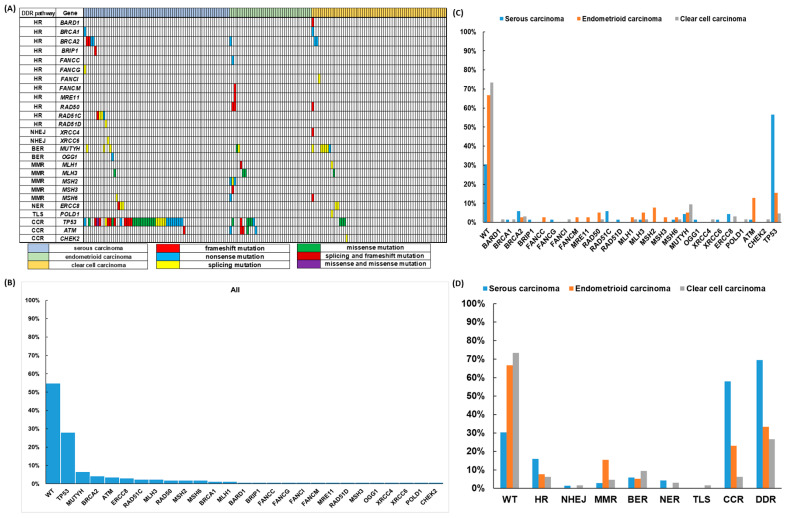
Deleterious DNA damage response (DDR) gene mutations in 172 epithelial ovarian carcinoma (EOC) patients (**A**) The pattern of DDR mutations of different histologic subtypes. (**B**) The percentages of DDR mutations in all 172 EOC patients. (**C**) The percentages of DDR mutations in different histologic subtypes. (**D**) The percentages of DDR mutations classified by different pathways in different histologic subtypes.

**Figure 2 biomedicines-09-01384-f002:**
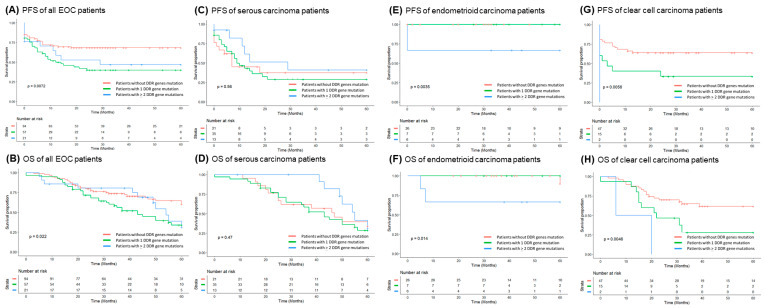
Kaplan–Meier analysis of progression-free survival (PFS) and overall survival (OS) in 172 epithelial ovarian carcinoma (EOC) patients. (**A**) PFS of 172 EOC patients. Note: EOC patients with DDR gene mutation(s) had shorter PFS (*p* = 0.0072, log-rank test). (**B**) OS of 172 EOC patients. Note: EOC patients with DDR gene mutation(s) had shorter OS (*p* = 0.022, log-rank test) (**C**) PFS of 69 serous carcinoma patients. Note: Serous carcinoma patients with ≥2 DDR gene mutations had a trend of better PFS although no statistical significance. (**D**) OS of 69 serous carcinoma patients. Note: Serous carcinoma patients with ≥2 DDR gene mutations had a trend of better OS although no statistical significance. (**E**) PFS of 39 endometrioid carcinoma patients. Note: Endometrioid carcinoma patients with ≥2 DDR gene mutations had poor PFS (*p* = 0.0035, log-rank test). (**F**) OS of 39 endometrioid carcinoma patients. Note: Endometrioid carcinoma patients with ≥2 DDR gene mutations had poor OS (*p* = 0.014, log-rank test). (**G**) PFS of 64 clear cell carcinoma patients. Note: Clear cell carcinoma patients with ≥2 DDR gene mutations had significantly shorter PFS (*p* = 0.0056, log-rank test). (**H**) OS of 64 clear cell carcinoma patients. Note: Clear cell carcinoma patients with ≥2 DDR gene mutations had significantly shorter OS (*p* = 0.0046, log-rank test).

**Table 1 biomedicines-09-01384-t001:** List of the DNA damage response (DDR) gene panel.

Gene	DDR Pathway	Gene	DDR Pathway
*ATM*	CCR	*ku70/XRCC6*	NHEJ
*BARD1*	HR	*ku80/XRCC5*	NHEJ
*BRCA1*	HR	*MDM4*	CCR
*BRCA2/FANCD1*	HR	*MLH1*	MMR
*BRIP1/FANCJ*	HR	*MLH3*	MMR
*CHEK2*	CCR	*MRE11*	HR
*DDB1*	NER	*MSH2*	MMR
*DDB2*	NER	*MSH3*	MMR
*ERCC1*	NER	*MSH6*	MMR
*ERCC2/XPD*	NER	*MUTYH*	BER
*ERCC3/XPB*	NER	*NBN*	HR
*ERCC4*	NER	*NBS1*	HR
*ERCC5/BIVM*	NER	*OGG1*	BER
*ERCC6/CSB*	NER	*PMS1*	MMR
*ERCC8/CSA*	NER	*PMS2*	MMR
*FANCA*	HR	*POLD1*	TLS
*FANCB*	HR	*POLE*	TLS
*FANCC*	HR	*POLB*	TLS
*FANCD1/BRCA2*	HR	*POLH*	TLS
*FANCD2*	HR	*POLK*	TLS
*FANCE*	HR	*RAD50*	HR
*FANCF*	HR	*RAD51*	HR
*FANCG/XRCC*	HR	*RAD51C/FANCO*	HR
*FANCI*	HR	*RAD51D*	HR
*FANCJ/BRIP1*	HR	*TP53*	CCR
*FANCL/PHF9*	HR	*XPA*	NER
*FANCM*	HR	*XPC*	NER
*FANCN/PALB2*	HR	*XRCC2*	NHEJ
*FANCO/RAD51C*	HR	*XRCC3*	NHEJ
*FANCP/SLX4*	HR	*XRCC4*	NHEJ

Note: BER: base excision repair; CCR: cell cycle regulation; DDR: DNA damage repair; HR: homologous recombination; MMR: mismatch repair; NER: nucleotide excision repair; NHEJ: nonhomologous DNA end joining; TLS: translesion synthesis.

**Table 2 biomedicines-09-01384-t002:** Characteristics of the epithelial ovarian cancer patients.

Patient Numbers	172
**Median Age (years old)**	52 (29–85)
**Median CA 125 (U/mL)**	400 (12–7265)
**Histology**	
Serous carcinoma	69 (40.1%)
Endometrioid carcinoma	39 (22.7%)
Clear cell carcinoma	64 (37.2%)
**FIGO stage**	
Early	69 (40.1%)
Advanced	103 (59.9%)
**Grade**	
Low	29 (16.9%)
High	143 (83.1%)
**Debulking surgery**	
Optimal	112 (65.1%)
Suboptimal	60 (34.9%)
**Recurrence**	
Yes	102 (59.3%)
No	70 (40.7%)
**Death**	
Yes	76 (44.2%)
No	96 (55.8%)

**Table 3 biomedicines-09-01384-t003:** The deleterious DDR gene mutations in the patients.

Gene	Mutation	Transcript	gDNA/cDNA	Amino Acid	Reported/Novel
*ATM*	frameshift deletion	NM_000051	c.1402_1403del	p.K468fs	rs587781347
*ATM*	frameshift deletion	NM_000051	c.8426delA	p.Q2809fs	rs587782558
*ATM*	frameshift insertion	NM_000051	c.4736dupA	p.Q1579fs	rs864622164
*ATM*	missense mutation	NM_000051	c.C6200A	p.A2067D	rs397514577
*ATM*	nonsense mutation	NM_000051	c.C5188T	p.R1730X	rs764389018
*ATM*	nonsense mutation	NM_000051	c.C850T	p.Q284X	rs757782702
*BARD1*	frameshift insertion	NM_000465	c.70_71insGT	p.P24fs	NA
*BRCA1*	nonsense mutation	NM_007294	c.3531dupT	p.S1178_K1179delinsX	NA
*BRCA1*	nonsense mutation	NM_007294	c.G2635T	p.E879X	rs80357251
*BRCA2*	frameshift deletion	NM_000059	c.1585delT	p.F529fs	NA
*BRCA2*	frameshift insertion	NM_000059	c.7407dupT	p.T2469fs	rs397507916
*BRCA2*	nonsense mutation	NM_000059	c.4965delC	p.Y1655X	rs80359475
*BRCA2*	nonsense mutation	NM_000059	c.A5623T	p.K1875X	NA
*BRCA2*	nonsense mutation	NM_000059	c.C2590T	p.Q864X	rs1060502414
*BRCA2*	nonsense mutation	NM_000059	c.C6952T	p.R2318X	rs80358920
*BRCA2*	nonsense mutation	NM_000059	c.G3922T	p.E1308X	rs80358638
*BRIP1*	frameshift insertion	NM_032043	c.394dupA	p.T132fs	rs587781416
*CHEK2*	splicing	NM_007194	g. 29130716 C>G		NA
*ERCC8*	frameshift deletion	NM_000082	c.191_195del	p.S64fs	NA
*ERCC8*	splicing	NM_000082	c.1123-2->T		NA
*ERCC8*	splicing	NM_000082	c.1123-2->T		NA
*ERCC8*	splicing	NM_000082	c.1123-2->T		NA
*ERCC8*	splicing	NM_000082	c.1123-2->T		rs777444521
*FANCC*	nonsense mutation	NM_000136	c.G1225T	p.E409X	NA
*FANCG*	splicing	NM_004629	c.511-2->C		NA
*FANCI*	splicing	NM_001113378	c.3187-2A>G		NA
*FANCM*	frameshift deletion	NM_020937	c.3998delA	p.Q1333fs	rs746983128
*MLH1*	frameshift deletion	NM_000249	c.1771delG	p.D591fs	NA
*MLH1*	splicing	NM_000249	c.2104-2A>G		rs267607889
*MLH1*	splicing	NM_000249	c.790+2T>C		rs267607790
*MLH3*	missense mutation	NM_001040108	c.G2221T	p.V741F	rs28756990
*MLH3*	missense mutation	NM_001040108	c.G2221T	p.V741F	rs28756990
*MLH3*	missense mutation	NM_001040108	c.G2221T	p.V741F	rs28756990
*MLH3*	missense mutation	NM_001040108	c.G2221T	p.V741F	rs28756990
*MRE11*	frameshift insertion	NM_005590	c.1222dupA	p.T408fs	rs774440500
*MSH2*	nonsense mutation	NM_000251	c.C226T	p.Q76X	rs63750042
*MSH2*	nonsense mutation	NM_000251	c.G1738T	p.E580X	rs63751411
*MSH2*	splicing	NM_000251	c.943-1G>C		rs12476364
*MSH3*	frameshift deletion	NM_002439	c.1141delA	p.K381fs	rs587776701
*MSH6*	frameshift insertion	NM_001281492	c.2916dupT	p.T972fs	NA
*MSH6*	nonsense mutation	NM_001281492	c.G726A	p.W242X	NA
*MSH6*	splicing	NM_001281492	g. 48033792 _ 48033795 del TAAC		NA
*MUTYH*	missense mutation	NM_001128425	c.G1187A	p.G396D	rs36053993
*MUTYH*	nonsense mutation	NM_001128425	c.G467A	p.W156X	rs762307622
*MUTYH*	splicing	NM_001128425	c.576+1G>C		NA
*MUTYH*	splicing	NM_001128425	c.934-2A>G		rs77542170
*MUTYH*	splicing	NM_001128425	c.934-2A>G		rs77542170
*MUTYH*	splicing	NM_001128425	c.934-2A>G		rs77542170
*MUTYH*	splicing	NM_001128425	c.934-2A>G		rs77542170
*MUTYH*	splicing	NM_001128425	c.934-2A>G		rs77542170
*MUTYH*	splicing	NM_001128425	c.934-2A>G		rs77542170
*MUTYH*	splicing	NM_001128425	c.934-2A>G		rs77542170
*MUTYH*	splicing	NM_001128425	c.934-2A>G		rs77542170
*OGG1*	nonsense mutation	NM_016819	c.A974G	p.X325W	NA
*POLD1*	splicing	NM_002691	c.2954-1G>-		NA
*RAD50*	frameshift deletion	NM_005732	c.2157delA	p.L719fs	NA
*RAD50*	frameshift deletion	NM_005732	c.536delT	p.I179fs	NA
*RAD50*	frameshift insertion	NM_005732	exon13:c.2157dupA	p.L719fs	rs397507178
*RAD51C*	frameshift insertion	NM_058216	c.390dupA	p.G130fs	rs730881940
*RAD51C*	nonsense mutation	NM_058216	c.T833G	p.L278X	NA
*RAD51C*	splicing	NM_058216	c.905-2A>C		NA
*RAD51C*	splicing	NM_058216	c.905-2A>C		NA
*RAD51D*	splicing	NM_002878	c.480+1G>A		NA
*TP53*	frameshift deletion	NM_000546	c.102delC	p.P34fs	NA
*TP53*	frameshift deletion	NM_000546	c.121delG	p.D41fs	NA
*TP53*	frameshift deletion	NM_000546	c.216delC	p.P72fs	NA
*TP53*	frameshift deletion	NM_000546	c.257_272del	p.A86fs	NA
*TP53*	frameshift deletion	NM_000546	c.501delG	p.Q167fs	NA
*TP53*	frameshift deletion	NM_000546	c.539_549del	p.E180fs	NA
*TP53*	frameshift insertion	NM_000546	c.102dupC	p.L35fs	NA
*TP53*	frameshift insertion	NM_000546	c.455dupC	p.P152fs	NA
*TP53*	frameshift insertion	NM_000546	c.498dupA	p.Q167fs	NA
*TP53*	frameshift insertion	NM_000546	c.889dupC	p.H297fs	NA
*TP53*	missense mutation	NM_000546	c.A488G	p.Y163C	rs148924904
*TP53*	missense mutation	NM_000546	c.A578G	p.H193R	rs786201838
*TP53*	missense mutation	NM_000546	c.A659C	p.Y220S	rs121912666
*TP53*	missense mutation	NM_000546	c.A659G	p.Y220C	rs121912666
*TP53*	missense mutation	NM_000546	c.A659G	p.Y220C	rs121912666
*TP53*	missense mutation	NM_000546	c.A736G	p.M246V	rs483352695
*TP53*	missense mutation	NM_000546	c.A838G	p.R280G	rs753660142
*TP53*	missense mutation	NM_000546	c.C380T	p.S127F	rs730881999
*TP53*	missense mutation	NM_000546	c.C451T	p.P151S	rs28934874
*TP53*	missense mutation	NM_000546	c.C844T	p.R282W	rs28934574
*TP53*	missense mutation	NM_000546	c.C844T	p.R282W	rs28934574
*TP53*	missense mutation	NM_000546	c.G412C	p.A138P	rs28934875
*TP53*	missense mutation	NM_000546	c.G524A	p.R175H	rs28934578
*TP53*	missense mutation	NM_000546	c.G524A	p.R175H	rs28934578
*TP53*	missense mutation	NM_000546	c.G638T	p.R213L	rs587778720
*TP53*	missense mutation	NM_000546	c.G730A	p.G244S	rs1057519989
*TP53*	missense mutation	NM_000546	c.G743A	p.R248Q	rs11540652
*TP53*	missense mutation	NM_000546	c.G743A	p.R248Q	rs11540652
*TP53*	missense mutation	NM_000546	c.G818A	p.R273H	rs28934576
*TP53*	missense mutation	NM_000546	c.G818A	p.R273H	rs28934576
*TP53*	missense mutation	NM_000546	c.G836A	p.G279E	rs1064793881
*TP53*	missense mutation	NM_000546	c.G856A	p.E286K	rs786201059
*TP53*	nonsense mutation	NM_000546	c.588_589insTGA	p.V197delinsX	NA
*TP53*	nonsense mutation	NM_000546	c.912dupT	p.K305_R306delinsX	NA
*TP53*	nonsense mutation	NM_000546	c.C430T	p.Q144X	NA
*TP53*	nonsense mutation	NM_000546	c.C499T	p.Q167X	NA
*TP53*	nonsense mutation	NM_000546	c.C574T	p.Q192X	NA
*TP53*	nonsense mutation	NM_000546	c.C586T	p.R196X	rs397516435
*TP53*	nonsense mutation	NM_000546	c.C637T	p.R213X	rs397516436
*TP53*	nonsense mutation	NM_000546	c.G272A	p.W91X	NA
*TP53*	nonsense mutation	NM_000546	c.G438A	p.W146X	NA
*TP53*	nonsense mutation	NM_000546	c.G859T	p.E287X	NA
*TP53*	nonsense mutation	NM_000546	c.G880T	p.E294X	rs1057520607
*TP53*	splicing	NM_000546	c.376-1G>T		NA
*TP53*	splicing	NM_000546	c.672+1G>A		rs863224499
*TP53*	splicing	NM_000546	c.993+2T>G		NA
*TP53*	splicing	NM_000546	c.993+2T>G		NA
*TP53*	splicing	NM_000546	c.993+1G>T		rs11575997
*TP53*	splicing	NM_000546	g.7577493_7577497 del CCTGA		NA
*XRCC4*	frameshift deletion	NM_003401	c.810delA	p.R270fs	NA
*XRCC6*	splicing	NM_001469	c.589+1G>T		NA

**Table 4 biomedicines-09-01384-t004:** The correlation of DDR gene mutations with clinical parameters in the epithelial ovarian cancer patients.

Genes		Histology			Type		FIGO Stage		Tumor Grade		Recurrence		Death	
		OSA	OEA	OCCA	I	II	Early	Advanced	Low	High	No	Yes	No	Yes
**Total**	172	69	39	64	104	68	69	103	29	143	70	102	96	76
**HR**														
**Wild type**	154	58	36	60	97	57	64	90	26	128	64	90	89	65
	89.53%	84.06%	92.31%	93.75%	93.27%	83.82%	92.75%	87.38%	89.66%	89.51%	91.43%	88.24%	92.71%	85.53%
**Mutation**	18	11	3	4	7	11	5	13	3	15	6	12	7	11
	10.47%	15.94%	7.69%	6.25%	6.73%	16.18%	7.25%	12.62%	10.34%	10.49%	8.57%	11.76%	7.29%	14.47%
***p* value ***				0.154		0.048		0.259		0.981		0.502		0.126
**NHEJ**														
**Wild type**	170	68	39	63	103	67	69	101	29	141	70	100	96	74
	98.84%	98.55%	100.00%	98.44%	99.04%	98.53%	100.00%	98.06%	100.00%	98.60%	100.00%	98.04%	100.00%	97.37%
**Mutation**	2	1	0	1	1	1	0	2	0	2	0	2	0	2
	1.16%	1.45%	0.00%	1.56%	0.96%	1.47%	0.00%	1.94%	0.00%	1.40%	0.00%	1.96%	0.00%	2.63%
***p* value***				0.742		0.761		0.244		0.522		0.239		0.11
**MMR**														
**Wild type**	161	67	33	61	95	66	65	96	24	137	66	95	91	70
	93.60%	97.10%	84.62%	95.31%	91.35%	97.06%	94.20%	93.20%	82.76%	95.80%	94.29%	93.14%	94.79%	92.11%
**Mutation**	11	2	6	3	9	2	4	7	5	6	4	7	5	6
	6.40%	2.90%	15.38%	4.69%	8.65%	2.94%	5.80%	6.80%	17.24%	4.20%	5.71%	6.86%	5.21%	7.89%
***p* value ***				0.03		0.134		0.793		0.009		0.762		0.475
**BER**														
**Wild type**	160	65	37	58	96	64	65	95	27	133	66	94	91	69
	93.02%	94.20%	94.87%	90.63%	92.31%	94.12%	94.20%	92.23%	93.10%	93.01%	94.29%	92.16%	94.79%	90.79%
**Mutation**	12	4	2	6	8	4	4	8	2	10	4	8	5	7
	6.98%	5.80%	5.13%	9.38%	7.69%	5.88%	5.80%	7.77%	6.90%	6.99%	5.71%	7.84%	5.21%	9.21%
***p* value ***				0.631		0.649		0.619		0.985		0.59		0.306
**NER**														
**Wild type**	167	66	39	62	102	65	67	100	29	138	67	100	93	74
	97.09%	95.65%	100.00%	96.88%	98.08%	95.59%	97.10%	97.09%	100.00%	96.50%	95.71%	98.04%	96.88%	97.37%
**Mutation**	5	3	0	2	2	3	2	3	0	5	3	2	3	2
	2.91%	4.35%	0.00%	3.13%	1.92%	4.41%	2.90%	2.91%	0.00%	3.50%	4.29%	1.96%	3.13%	2.63%
***p* value ***				0.43		0.342		0.996		0.307		0.373		0.848
**TLS**														
**Wild type**	171	69	39	63	103	68	69	102	29	142	70	101	96	75
	99.42%	100.00%	100.00%	98.44%	99.04%	100.00%	100.00%	99.03%	100.00%	99.30%	100.00%	99.02%	100.00%	98.68%
**Mutation**	1	0	0	1	1	0	0	1	0	1	0	1	0	1
	0.58%	0.00%	0.00%	1.56%	0.96%	0.00%	0.00%	0.97%	0.00%	0.70%	0.00%	0.98%	0.00%	1.32%
***p* value ***				0.428		0.417		0.412		0.652		0.406		0.26
**DSBR**														
**Wild type**	153	57	36	60	97	56	64	89	26	127	64	89	89	64
	88.95%	82.61%	92.31%	93.75%	93.27%	82.35%	92.75%	86.41%	89.66%	88.81%	91.43%	87.25%	92.71%	84.21%
**Mutation**	19	12	3	4	7	12	5	14	3	16	6	13	7	12
	11.05%	17.39%	7.69%	6.25%	6.73%	17.65%	7.25%	13.59%	10.34%	11.19%	8.57%	12.75%	7.29%	15.79%
***p* value ***				0.092		0.026		0.193		0.895		0.391		0.077
**SSBR**														
**Wild type**	145	60	31	54	86	59	59	86	22	123	59	86	83	62
	84.30%	86.96%	79.49%	84.38%	82.69%	86.76%	85.51%	83.50%	75.86%	86.01%	84.29%	84.31%	86.46%	81.58%
**Mutation**	27	9	8	10	18	9	10	17	7	20	11	16	13	14
	15.70%	13.04%	20.51%	15.63%	17.31%	13.24%	14.49%	16.50%	24.14%	13.99%	15.71%	15.69%	13.54%	18.42%
***p* value ***				0.591		0.473		0.722		0.171		0.996		0.382
**CCR**														
**Wild type**	119	29	30	60	91	28	60	59	24	95	57	62	74	45
	69.19%	42.03%	76.92%	93.75%	87.50%	41.18%	86.96%	57.28%	82.76%	66.43%	81.43%	60.78%	77.08%	59.21%
**Mutation**	53	40	9	4	13	40	9	44	5	48	13	40	22	31
	30.81%	57.97%	23.08%	6.25%	12.50%	58.82%	13.04%	42.72%	17.24%	33.57%	18.57%	39.22%	22.92%	40.79%
***p* value ***				<0.001		<0.001		<0.001		0.083		0.004		0.012
**DDR**														
**Wild type**	94	21	26	47	74	20	50	44	20	74	47	47	63	31
	54.65%	30.43%	66.67%	73.44%	71.15%	29.41%	72.46%	42.72%	68.97%	51.75%	67.14%	46.08%	65.63%	40.79%
**1 gene**	57	35	7	15	22	35	14	43	5	52	16	41	24	33
**mutation**	33.14%	50.72%	17.95%	23.44%	21.15%	51.47%	20.29%	41.75%	17.24%	36.36%	22.86%	40.20%	25.00%	43.42%
**2 gene**	15	12	2	1	3	12	2	13	0	15	4	11	6	9
**mutations**	8.72%	17.39%	5.13%	1.56%	2.88%	17.65%	2.90%	12.62%	0.00%	10.49%	5.71%	10.78%	6.25%	11.84%
**3 gene**	2	1	1	0	1	1	1	1	1	1	1	1	1	1
**mutations**	1.16%	1.45%	2.56%	0.00%	0.96%	1.47%	1.45%	0.97%	3.45%	0.70%	1.43%	0.98%	1.04%	1.32%
**4 gene**	2	0	2	0	2	0	2	0	2	0	2	0	2	0
**mutations**	1.16%	0.00%	5.13%	0.00%	1.92%	0.00%	2.90%	0.00%	6.90%	0.00%	2.86%	0.00%	2.08%	0.00%
**5 gene**	1	0	1	0	1	0	0	1	1	0	0	1	0	1
**Mutations**	0.58%	0.00%	2.56%	0.00%	0.96%	0.00%	0.00%	0.97%	3.45%	0.00%	0.00%	0.98%	0.00%	1.32%
**6 gene**	1	0	0	1	1	0	0	1	0	1	0	1	0	1
**mutations**	0.58%	0.00%	0.00%	1.56%	0.96%	0.00%	0.00%	0.97%	0.00%	0.70%	0.00%	0.98%	0.00%	1.32%
**Total**	78	48	13	17	30	48	19	59	9	69	23	55	33	45
**mutations**	45.35%	69.57%	33.33%	26.56%	28.85%	70.59%	27.54%	57.28%	31.03%	48.25%	32.86%	53.92%	34.38%	59.21%
***p* value ***				<0.001		<0.001		<0.001		0.089		0.006		0.001

Note: BER: base excision repair; CCR: cell cycle regulation; DDR: DNA damage response; DSBR: double-strand break repair; HR: homologous recombination; MMR: mismatch repair; NER: nucleotide excision repair; NHEJ: nonhomologous DNA end joining; OSA: ovarian serous carcinoma; OEA: ovarian endometrioid carcinoma; OCCA: ovarian clear cell carcinoma; SSBR: single-strand break repair; TLS: translesion synthesis. * Pearson’s chi-squared test

**Table 5 biomedicines-09-01384-t005:** Cox regression model for the risk factors for recurrence and death in all patients (*n* = 172).

Factors		Recurrence				Death			
		Univariate		Multivariate		Univariate		Multivariate	
	n	Hazard Ratio (95% CI)	*p*	Hazard Ratio (95% CI)	*p*	Hazard Ratio (95% CI)	*p*	Hazard Ratio (95% CI)	*p*
**Histology**									
**OSA**	69	1 (reference)		1 (reference)		1 (reference)		1 (reference)	
**OEA**	39	0.17 (0.08–0.37)	<0.001	0.42 (0.16–1.12)	0.082	0.12 (0.04–0.38)	<0.001	0.45 (0.13–1.55)	0.205
**OCCA**	64	0.96 (0.64–1.44)	0.835			1.37 (0.86–2.18)	0.188		
**Type**									
**I**	104	1 (reference)		1 (reference)		1 (reference)		1 (reference)	
**II**	68	2.69 (1.81–4.00)	<0.001	0.77 (0.46–1.28)	0.311	1.88 (1.19–2.96)	0.007	0.35 (0.20–0.60)	<0.001
**FIGO stage**									
**Early**	69	1 (reference)		1 (reference)		1 (reference)		1 (reference)	
**Advanced**	103	5.29 (3.16–8.85)	<0.001	3.08 (1.63–5.80)	0.001	6.84 (3.28–14.25)	<0.001	4.82 (2.09–11.09)	<0.001
**Tumor grade**									
**Low**	29	1 (reference)		1 (reference)		1 (reference)		1 (reference)	
**High**	143	5.57 (2.26–13.70)	<0.001	1.68 (0.55–5.15)	0.366	17.97 (2.50–129.29)	0.004	7.38 (0.93–58.28)	0.058
**Debulking surgery**									
**Suboptimal**	60	1 (reference)		1 (reference)		1 (reference)		1 (reference)	
**Optimal**	112	0.28 (0.18–0.41)	<0.001	0.51 (0.32–0.80)	0.004	0.26 (0.16–0.41)	<0.001	0.38 (0.22–0.64)	<0.001
**HR**									
**Wild type**	154	1 (reference)				1 (reference)			
**Mutation**	18	1.22 (0.67–2.23)	0.516			1.15 (0.59–2.25)	0.674		
**NHEJ**									
**Wild type**	170	1 (reference)				1 (reference)			
**Mutation**	2	2.04 (0.50–8.28)	0.319			2.52 (0.62–10.32)	0.197		
**MMR**									
**Wild type**	161	1 (reference)				1 (reference)			
**Mutation**	11	1.31 (0.61–2.83)	0.487			1.88 (0.81–4.33)	0.139		
**BER**									
**Wild type**	160	1 (reference)				1 (reference)			
**Mutation**	12	1.32 (0.64–2.71)	0.454			1.70 (0.78–3.72)	0.185		
**NER**									
**Wild type**	167	1 (reference)				1 (reference)			
**Mutation**	5	0.58 (0.14–2.36)	0.449			0.71 (0.18–2.91)	0.639		
**TLS**									
**Wild type**	171	1 (reference)				1 (reference)		1 (reference)	
**Mutation**	1	5.19 (0.71–37.89)	0.104			33.76 (3.95–289.00)	0.001	9.57 (1.08–84.83)	0.042
**DSBR**									
**Wild type**	153	1 (reference)				1 (reference)			
**Mutation**	19	1.23 (0.69–2.20)	0.488			1.20 (0.63–2.27)	0.584		
**SSBR**									
**Wild type**	145	1 (reference)				1 (reference)			
**Mutation**	27	1.10 (0.64–1.87)	0.736			1.46 (0.82–2.61)	0.202		
**CCR**									
**Wild type**	119	1 (reference)		1 (reference)		1 (reference)			
**Mutation**	53	1.68 (1.12–2.50)	0.011	0.98 (0.58–1.66)	0.939	1.54 (0.97–2.45)	0.066		
**DDR**									
**Wild type**	94	1 (reference)		1 (reference)		1 (reference)		1 (reference)	
**1 gene mutation**	57	1.71 (1.12–2.60)	0.013	1.18 (0.73–1.91)	0.496	1.96 (1.20–3.20)	0.007	1.57 (0.97–2.54)	0.062
**≥2 gene mutations**	21	1.52 (0.84–2.76)	0.171			1.56 (0.78–3.11)	0.207		

Note: BER: base excision repair; CCR: cell cycle regulation; DDR: DNA damage response; DSBR: double-strand break repair;HR: homologous recombination; MMR: mismatch repair; NER: nucleotide excision repair; NHEJ: nonhomologous DNA end joining; OSA: ovarian serous carcinoma; OEA: ovarian endometrioid carcinoma; OCCA: ovarian clear cell carcinoma; SSBR: single-strand break repair; TLS: translesion synthesis.

## Data Availability

The datasets generated and/or analyzed during the current study are available from the corresponding author on reasonable request.

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
