# Peer review of "A DNA Damage Response Gene Panel for Different Histologic Types of Epithelial Ovarian Carcinomas and Their Outcomes"

_biomedicines, 2021, doi:10.3390/biomedicines9101384_

Round 1
Reviewer 1 Report
It is not clear how many of the patients are high grade serous ovarian cancer (HGSOC). It is important to also perform the different analysis on just HGSOC and not include low grade serous in analyses. It would also be useful to compare the type I and type II ovarian cancer subtypes.
Author Response
Response 1: We appreciate the suggestion and perform the analysis in the revised manuscript. In our cohort, there were 68 patients of type II tumors (HGSOC) and 104 type I tumors. We compare the type I and type II ovarian cancer subtypes in the revised manuscript (Please see line 41-42, 171-172, 207-230, 249-265, 294-297, 307-310, 364-365, Table 4 and Table 5).
Response 2: We appreciate the suggestion and our manuscript has undergone English language editing by MDPI.
Reviewer 2 Report
The authors study the correlation between different mutations in DDR genes and clinico-pathologyc characteristics of EOC. They found alterations in 26 of the 60 genes analyzed (from seven DDR pathways). A correlation between this alterations and unfavorable scenarios was found. Moreover, they show the prognostic value of these genetics’ alterations.
The authors say that “the recurrent patients had higher percentage of DDR genes mutation than those without recurrence (lines 212, 213). However, for the survival analyses they only consider presence (at least one) or absence of mutations. I think that the influence of the accumulation of mutations should be studied in the survival studies. For instance, the multivariate analysis show that the suggested variable (DDR wild type or mutation) has no independent value. Once again, I think that a detailed study about the importance of each one of DDR pathways analyzed would offer results with greater clinical relevance. In my opinion, this work would improve notably if this type of analysis were included.
Correct “Table 69” in line 252.
I think it is more appropriate the use of “progression free survival” than “disease free survival”, since residual tumors are present in many patients after surgery.
The authors study the correlation between different mutations in DDR genes and clinico-pathologyc characteristics of EOC. They found alterations in 26 of the 60 genes analyzed (from seven DDR pathways). A correlation between this alterations and unfavorable scenarios was found. Moreover, they show the prognostic value of these genetics’ alterations.
The authors say that “the recurrent patients had higher percentage of DDR genes mutation than those without recurrence (lines 212, 213). However, for the survival analyses they only consider presence (at least one) or absence of mutations. I think that the influence of the accumulation of mutations should be studied in the survival studies. For instance, the multivariate analysis show that the suggested variable (DDR wild type or mutation) has no independent value. Once again, I think that a detailed study about the importance of each one of DDR pathways analyzed would offer results with greater clinical relevance. In my opinion, this work would improve notably if this type of analysis were included.
Correct “Table 69” in line 252.
I think it is more appropriate the use of “progression free survival” than “disease free survival”, since residual tumors are present in many patients after surgery.
Author Response
Point 1: The authors study the correlation between different mutations in DDR genes and clinico-pathologyc characteristics of EOC. They found alterations in 26 of the 60 genes analyzed (from seven DDR pathways). A correlation between this alterations and unfavorable scenarios was found. Moreover, they show the prognostic value of these genetics’ alterations. The authors say that “the recurrent patients had higher percentage of DDR genes mutation than those without recurrence (lines 212, 213). However, for the survival analyses they only consider presence (at least one) or absence of mutations. I think that the influence of the accumulation of mutations should be studied in the survival studies. For instance, the multivariate analysis show that the suggested variable (DDR wild type or mutation) has no independent value. Once again, I think that a detailed study about the importance of each one of DDR pathways analyzed would offer results with greater clinical relevance. In my opinion, this work would improve notably if this type of analysis were included.
Response 1: We appreciate the suggestion and investigate the influence of the accumulation of mutations in the survival studies in the revised manuscript (Please see line 238-265, Figure 2, Table 5). We also perform the analysis of each one of DDR pathways in the revised manuscript (Please see line 249-265, 357-359, Table 4 and Table 5)
Point 2: Correct “Table 69” in line 252.
Response 2: We appreciate the suggestion, and we corrected it in the revised manuscript (Please see line 270).
Point 3: I think it is more appropriate the use of “progression free survival” than “disease free survival”, since residual tumors are present in many patients after surgery.
Response 3: We appreciate the suggestion, and we revised it in the revised manuscript. (Please see line 106-107, 166, 238, 268 and Figure 2)
Round 2
Reviewer 1 Report
The authors addressed my comments